# Temporal convolutional network for a Fast DNA mutation detection in breast cancer data

Untari Novia Wisesty[1,2], Tati Rajab Mengko[3]*, Ayu Purwarianti[3,4], Adi Pancoro[5]

1 Bandung Institute of Technology, Doctoral Program of Electrical Engineering and Informatics, School of Electrical and Information Engineering, Bandung, Indonesia, 2 School of Computing, Telkom University, Bandung, Indonesia, 3 Bandung Institute of Technology, School of Electrical and Information Engineering, Bandung, Indonesia, 4 U-CoE AI-VLB, Bandung, Indonesia, 5 Bandung Institute of Technology, School of Life Sciences and Technology, Bandung, Indonesia

* tatirajabmengko@gmail.com

**Data Availability Statement:** The COSMIC dataset are available from website https://cancer.sanger.ac.uk/cosmic, and RSCM dataset in Fastq raw file are available from Sequence Read Archive accession numbers SRR11774092–SRR11774115.

## Abstract

Early detection of breast cancer can be achieved through mutation detection in DNA sequences, which can be acquired through patient blood samples. Mutation detection can be performed using alignment and machine learning techniques. However, alignment techniques require reference sequences, and machine learning techniques still cannot predict index mutation and require supporting tools. Therefore, in this research, a Temporal Convolutional Network (TCN) model was proposed to detect the type and index mutation faster and without reference sequences and supporting tools. The architecture of the proposed TCN model is specifically designed for sequential labeling tasks on DNA sequence data. This allows for the detection of the mutation type of each nucleotide in the sequence, and if the nucleotide has a mutation, the index mutation can be obtained. The proposed model also uses 2-mers and 3-mers mapping techniques to improve detection performance. Based on the tests that have been carried out, the proposed TCN model can achieve the highest F1-score of 0.9443 for COSMIC dataset and 0.9629 for RSCM dataset, Additionally, the proposed TCN model can detect index mutation six times faster than BiLSTM model. Furthermore, the proposed model can detect type and index mutations based on the patient's DNA sequence, without the need for reference sequences or other additional tools.

## Introduction

Breast cancer is the primary cancer that affects women, although men can also suffer from this disease. According to data from the Global Cancer Observatory (GLOBOCAN), it was estimated that the number of new cases would increase to 19.3 million in 2020, resulting 10 million deaths caused by cancer in [1]. Of the new cases, 24.5% of cases were breast cancer experienced by women, indicating that breast cancer is the disease with the highest number of cases among to other cancers suffered by women. In Indonesia, there were 65,858 new cases of breast cancer in 2020 with a mortality rate of 9.6%. Breast cancer is the second most deadly cancer after lung cancer. This high mortality rate can be attributed to the disease not being treated, perhaps due to detection at a later stage, resulting in lower treatment efficacy.

**Funding:** A part of this research is funded by 2022 Research, Community Service and Innovation Program, School of Electrical and Information Engineering, Bandung Institute of Technology.

**Competing interests:** The authors have declared that no competing interests exist.

Therefore, early detection of breast cancer is crucial to reducing the mortality rate caused by breast cancer and increasing the success rate of treatment. Diagnostic mammograms, breast ultrasound, and breast magnetic resonance imaging (MRI) can be used to detect breast cancer [2], but these tests can only be performed if a tumor has already appeared in the breast.

Based on several studies that have been carried out, one of the causes of the increased risk of breast cancer is abnormalities in the patient's DNA. Such abnormalities can be mutations or changes in nucleotides in certain genes. The most common genes that correlate with breast cancer are the BRCA1 and BRCA2 genes [3]. The BRCA1 and BRCA2 genes function as tumor suppressors. If there is a mutation in one of the genes, it can cause genomic instability, as well as cell changes from normal cells to cancer cells [4]. Other genes that can cause an increased risk of breast cancer were also mentioned in other studies, including PTEN, TP53, CDH1, STK11, CHEK2, ATM, BRIP1, PALB2, RAD51D, BARD1, RAD51C, NBN, and XRCC2. A person who has mutations in these genes can pass them on to their offspring, so their offspring will have a higher risk of developing breast cancer [3, 5–7]. Furthermore, early detection of the risk of breast cancer can be done by detecting DNA mutations in certain genes through patient blood samples. This makes the examination safer to perform, and patients are more aware of maintaining their health if there are indications of a risk of developing breast cancer.

In the field of bioinformatics, two approaches are generally used to detect DNA mutations, namely the alignment approach and machine learning approaches. Several studies have been developed using the alignment approach, such as AutoCSA [8], IsoMut [9], sweep-based seed processing [10], MutClustSW [11], and Bivartect [12]. The main problems that occur when using the alignment approach are the requirement of reference sequences and the need for a long test time to detect mutations in new sequences because the new sequences must be matched with all relevant reference sequences.

The second approach in detecting mutations using DNA sequence data is the machine learning approach. This approach has the advantage that the time required to test new samples is relatively faster than the alignment approach, because the new samples to be tested are entered into the model without being matched one by one to the relevant reference samples. Several studies have used the machine learning approach, namely SNVHMM [13], Bayesian based Somatic Structural Variation [14], Functional Link Artificial Neural Network [15], Tumor Covariate Signature Model [16], DriverML [17], Multi-Modal Deep Boltzmann Machine [18], NeoSomatic [19], circulating tumor DNA detection [20].

The machine learning approach has several drawbacks, including the lower level of accuracy in mutation detection when compared to alignment techniques. Additionally, most existing research only classifies whether a sequence has mutations or not, without predicting where the mutation occurred. In our previous research, we used the Bidirectional Long Short-Term Memory (BiLSTM) model to detect mutations in DNA sequences [21]. However, the BiLSTM still requires some time to detect mutations, and the data used is limited to single gene datasets in lung cancer acquired from the Catalog of Somatic Mutations in Cancer (COSMIC database) [22].

In this research, a TCN model was proposed to detect genetic mutations along with the mutation index found in the inputted DNA sequence. The input needed for the proposed research is only in the form of a DNA sequence consisting of nucleotide sequences A, C, T, and G, and without a reference sequence at the time of mutation detection, making it easier for users to use the resulting model. The method used is one of the Deep Learning methods, known as Temporal Convolutional Network (TCN). The Deep Learning model can process data end to end, so it does not require additional methods at the feature selection stage. It has been widely used to solve problems, especially in the biomedical field [23–25]. Furthermore,

TCN is one of the developments of the Convolutional Neural Network (CNN), which is suitable for use in sequential data and has large receptive fields. In several studies, TCN has managed to achieve excellent performance, including detecting El Nino-Southern Oscillation [26], anomaly detection in time series data [27], action segmentation [28, 29], time series data classification [30], short-term prediction of wind strength [31], and speech and music detection [32]. Bai et al [33] conducted benchmark tests on several standard sequence modelling tasks, which included adding problems, sequential MNIST and P-MNIST, copy memory, JSB Chorales and Nottingham, PennTreebank, Wikitext-103, LAMBADA, and text8 [34–39]. From several tests conducted, it can be concluded that TCN has better performance than some variations of the Recurrent Neural Network (RNN) method in most sequence modelling tasks tested.

The novelty proposed in this paper includes the detection of type and index mutation in DNA sequence data for breast cancer patients using the TCN model. The data inputted is only in the form of DNA sequences and without using reference sequences or other additional tools. The first dataset used was obtained from the public database, namely the Catalogue of Somatic Mutations in Cancer (COSMIC dataset) [22], which includes a combination of the genes ATM, BARD1, BRCA1, BRCA2, BRIP1, CDH1, CHEK2, EPCAM, MLH1, MRE11, MSH2, MSH6, NBN, PALB2, PMS2, PTEN, RAD50, RAD51C, RAD51D, TP53, and XRCC2. Additionally, this research, also use DNA sequence data from breast cancer patients at Cipto Mangunkusumo National Hospital, Jakarta, Indonesia (RSCM dataset) [40]. The data is in the form of sequencing results from patient blood samples in the form of short sequences and have a high degree of heterogeneity and with a larger number of sequences.

The architecture of the proposed TCN model is specifically designed for sequential labeling tasks on DNA sequence data. This allows for the detection of the mutation type of each nucleotide in the sequence, and if the nucleotide has a mutation, the index mutation can be obtained. The proposed model can also detect one or more mutations in one input sequence along with the index where the mutation occurs with a faster detection time. 2-mers and 3-mers mapping techniques are also proposed in this research to improve the performance level in detecting type and index mutation. The proposed TCN model succeeded in achieving an F1-score of 0.9427 for the COSMIC dataset and 0.9629 for the RSCM dataset, and was six times faster than BiLSTM model. Fig 1 shows the advantages and research gap between the proposed TCN model and the related work. Finnaly, in this research, all authors planned and conceptualized the conducted research, U.W. carried out the implementation and experimentation of the proposed model, analysis the results, and wrote the draft paper, T.M. led and supervised the project, while A.P. and A.P. analyzed the results and proofread the manuscript.

## Materials and methods

In this section, a detailed explanation of the dataset and proposed methods for detecting index mutation in the DNA sequence of breast cancer patients is presented. Two datasets are used, including a dataset acquired from the COSMIC database consisting of a combination of 21 genes related to breast cancer (COSMIC dataset) and DNA sequence dataset of breast cancer patients from Cipto Mangunkusumo National Hospital, Jakarta, Indonesia (RSCM dataset). The dataset undergo a data preprocessing process, which includes DNA mapping, sequence padding, and sequence reshaping. Then, to detect the index mutation, a sequential labeling model using the TCN method is proposed. Fig 2 shows a block diagram for the proposed research.

**Alignment**
+ Can detect the type and index mutation from alignment results.
- Requires reference sequences.
- Requires large memory.
- Takes a long time to detect mutations.

**Machine Learning**
+ Faster mutation detection time.
- Only classify mutation or not.
- Still using other supporting data and tools.

**Proposed TCN Model**
+ Detect the type and index mutation.
+ Can detect one or more mutation along with its index in one sequence.
+ Only uses DNA sequence data.
+ Does not use reference sequences.
+ Faster mutation detection time.

**Fig 1. The advantages of the proposed TCN model compared to the related work.**

### Data acquisition and preprocessing

The DNA sequence dataset of breast cancer patients used in this research was obtained from two different sources, including the COSMIC database, which is a public database (COSMIC dataset), and the RSCM dataset, which is a DNA sequence dataset specifically for breast cancer patients in Indonesia. The RSCM dataset was obtained from Cipto Mangunkusumo National Hospital, Jakarta, Indonesia. All dataset used in this research were acquired from published

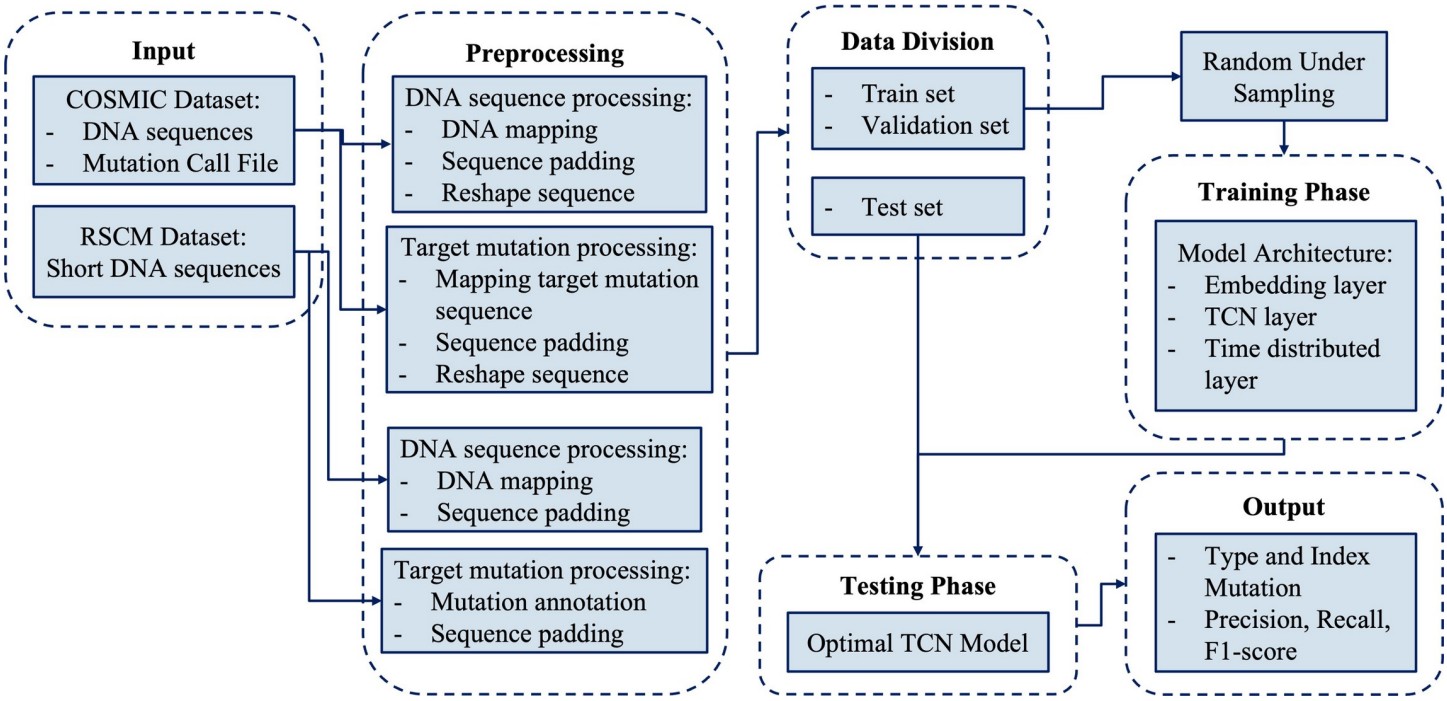

**Fig 2. Block diagram of the proposed index mutation detection using TCN model.**

data and analyzed anonymously. The first dataset is COSMIC dataset which is a combination of 21 gene datasets used when examining breast cancer based on Kailos' Heredity Cancer Screening Panel. The COSMIC dataset consists of ATM, BARD1, BRCA1, BRCA2, BRIP1, CDH1, CHEK2, EPCAM, MLH1, MRE11, MSH2, MSH6, NBN, PALB2, PMS2, PTEN, RAD50, RAD51C, RAD51D, TP53, and XRCC2 genes. The acquired data consists of two parts, namely mutation call data and DNA sequence reference data. Mutation call data includes gene name, sequence transcript, sample name, sample ID, AA mutation, CDS mutation (type and index DNA mutation), primary tissue, and others details. The data includes several types of mutations, namely substitution (SNV), insertion, deletion, and duplicate. Duplicate mutations are later combined with insertion mutations because they have an additional number of nucleotides, as in insertion mutation. Then, the data is processed by generating patient sequences (samples) based on mutation data that occurred in certain samples and corresponding reference sequences [21]. From the preprocessing process, a total of 81,272 samples were obtained with different types of genes and the number of mutations. Fig 3 shows the mutation rates of SNV, insertion, and deletion mutations in each gene. The occurrence percentage of SNV mutations is 76.33% of the total mutations that occur, 10.79% for insertion mutations, and 12.88% for deletion mutations.

The second dataset observed was the RSCM dataset from which samples were taken from patients at Cipto Mangunkusumo National Hospital, Jakarta, Indonesia. The dataset consists of 24 breast cancer samples/patients, and each sample consists of millions of short sequences for which the index mutation has not been annotated. Therefore, the initial stage for processing the RSCM dataset is to annotate mutations on each sequence. In this research, mutation annotations were performed based on alignment results using the multi-BLAST tool. Basic Local Alignment Search Tool (BLAST) is a bioinformatics tool that is widely known and used in many tasks in the field of bioinformatics. BLAST searches the local region of the query sequence and compares it with the nucleotide or protein sequences in the database [41, 42]. Alignment and sequence searches were performed towards the human genome reference GRCh38.p13 reference primary assembly top-level database. In addition to the alignment results, the output of multi-BLAST also provides information on the chromosomes where the query sequence is found and its first and last points. In the annotations carried out, normal sequences and sequences containing SNV mutations with a similarity level above 99% were taken with the human genome reference database used. Then, the annotated sequences were divided into three datasets containing 100,000 (100k), 250,000 (250k), and 500,000 (500k) of the first short sequence of each sample. Thus, the first dataset (100k) contains a total of 2,400,000 short sequences, 6,000,000 short sequences for the second dataset (250k), and 12,000,000 short sequences for the third dataset (500k). The formation of the three datasets aims to determine the influence of the amount of data on the TCN method used. Figs 4 and 5 show the comparison of SNV mutation rates on 100k, 250k, and 500k datasets for each sample and chromosome. Based on the graph, the more data used in each sample, the greater the number of SNVs that occur (Fig 4), and many SNV mutations occur on chromosome 20 (Fig 5).

Furthermore, the available DNA sequences are padded to equalize the length of the DNA sequence. Sequence padding is done by adding the character 'N' at the end of the sequence. DNA sequence mapping is carried out, which involves converting nucleotides in sequences into numerical sequences. This is done because the proposed TCN model requires numeric-type data input. In this research, several mapping techniques were used, including integer mapping, 2-mers mapping, and 3-mers mapping. Integer mapping converts one nucleotide into a single numerical value, where nucleotide T is converted into value 1, nucleotide C is converted into value 2, nucleotide A is converted into value 3, and nucleotide G is converted into value 4 [43, 44]. In the 2-mers mapping technique, two consecutive nucleotides are

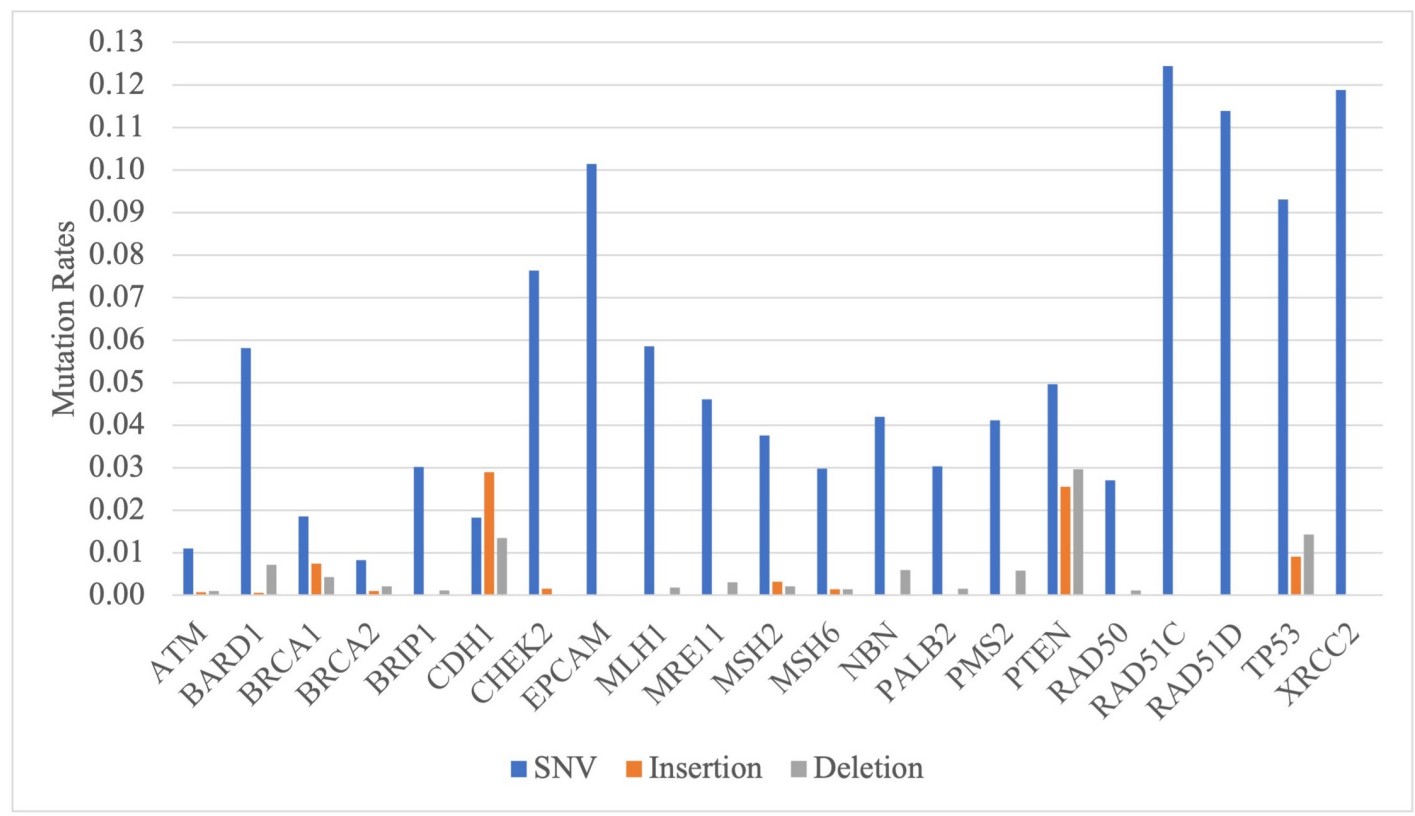

**Fig 3. Mutation rates in each gene on COSMIC dataset.**

converted into one numerical value, and in the 3-mers mapping technique, three consecutive nucleotides are converted into one numerical value. For example, in the 2-mers mapping

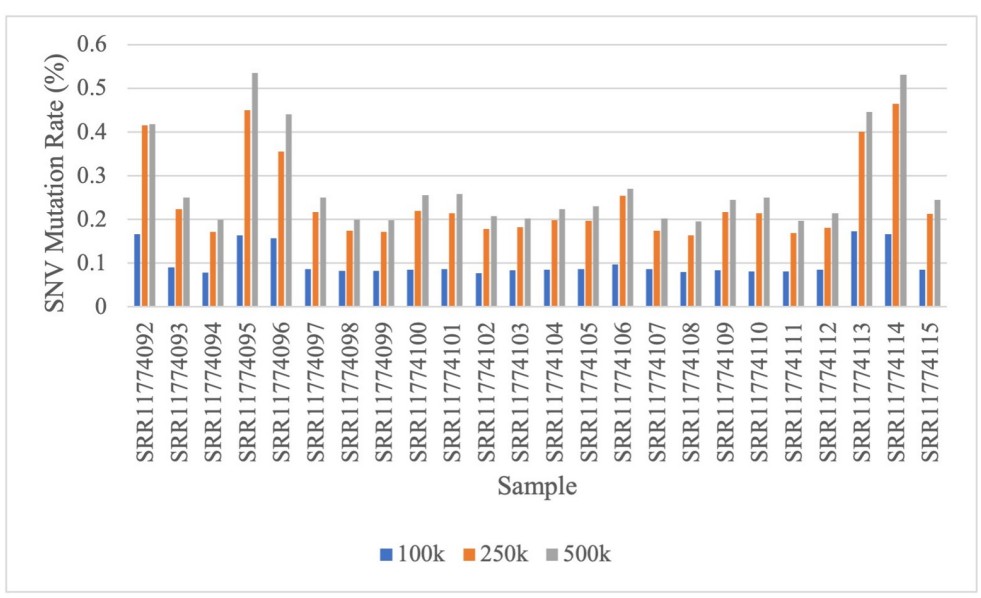

**Fig 4. SNV rates comparison on 100k, 250k, and 500k datasets for each sample on RSCM dataset.**

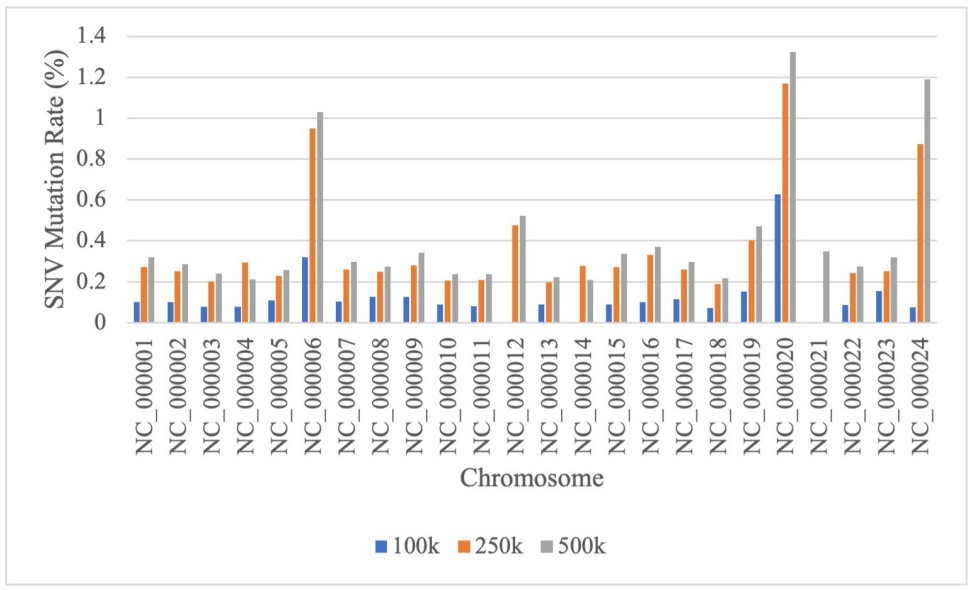

**Fig 5. SNV rates comparison on 100k, 250k, and 500k datasets for each chromosome on RSCM dataset.**

technique, 'AA' is converted into a value of 1, 'AT' becomes 2, and so on, while in the 3-mers mapping technique, 'AAA' is converted into a value of 1, 'AAT' becomes 2, and so on. Tables 1 and 2 show conversion tables for 2-mers and 3-mers mapping techniques. The padding character 'N' is converted to a value of 0. Furthermore, in the COSMIC dataset, DNA sequences that have been converted into numerical sequences are reshaped using a sliding window technique to shorten the length of the sequence. The length of the subsequence used is 150 bp with an overlap of 50 bp. Meanwhile, in the RSCM dataset, no sequence reshapeing is carried out because each sequence in the RSCM dataset has a maximum length of 144 bp.

The proposed mutation detection model uses a sequential labelling model in which each nucleotide in the sequence will be detected whether it has a mutation or not (normal nucleotide). Therefore, the actual mutation target/output is converted into a target sequence that is the same length as the DNA sequence input. Each value in the target sequence is labeled 0 if the nucleotide is normal/not mutated, 1 if the nucleotide has an SNV mutation, 2 if the nucleotide has an insertion mutation, and 3 if the nucleotide has a deletion mutation. The target sequence undergoes a process of sequence reshaping and sequence padding with the same process as DNA sequence input. Then each COSMIC dataset and the three RSCM datasets are divided into two parts, the train set and test set, in a 90:10 ratio. Especially for train sets, data sampling is carried out using the Random Under Sampling technique to balance data labels in each class by randomly reducing the number of majority class sequences, namely sequences that do not contain mutations. The train sets are later used in the TCN

**Table 1. 2-mers table conversion.**

| 2-mers | Numeric | 2-mers | Numeric | 2-mers | Numeric | 2-mers | Numeric |
|---|---|---|---|---|---|---|---|
| AA | 1 | AT | 2 | AG | 3 | AC | 4 |
| TA | 5 | TT | 6 | TG | 7 | TC | 8 |
| GA | 9 | GT | 10 | GG | 11 | GC | 12 |
| CA | 13 | CT | 14 | CG | 15 | CC | 16 |

**Table 2. 3-mers table conversion.**

| 3-mers | Numeric | 3-mers | Numeric | 3-mers | Numeric | 3-mers | Numeric |
|---|---|---|---|---|---|---|---|
| AAA | 1 | AAT | 2 | AAG | 3 | AAC | 4 |
| ATA | 5 | ATT | 6 | ATG | 7 | ATC | 8 |
| AGA | 9 | AGT | 10 | AGG | 11 | AGC | 12 |
| ACA | 13 | ACT | 14 | ACG | 15 | ACC | 16 |
| TAA | 17 | TAT | 18 | TAG | 19 | TAC | 20 |
| TTA | 21 | TTT | 22 | TTG | 23 | TTC | 24 |
| TGA | 25 | TGT | 26 | TGG | 27 | TGC | 28 |
| TCA | 29 | TCT | 30 | TCG | 31 | TCC | 32 |
| GAA | 33 | GAT | 34 | GAG | 35 | GAC | 36 |
| GTA | 37 | GTT | 38 | GTG | 39 | GTC | 40 |
| GGA | 41 | GGT | 42 | GGG | 43 | GGC | 44 |
| GCA | 45 | GCT | 46 | GCG | 47 | GCC | 48 |
| CAA | 49 | CAT | 50 | CAG | 51 | CAC | 52 |
| CTA | 53 | CTT | 54 | CTG | 55 | CTC | 56 |
| CGA | 57 | CGT | 58 | CGG | 59 | CGC | 60 |
| CCA | 61 | CCT | 62 | CCG | 63 | CCC | 64 |

model training process, while the test sets are used to measure system performance after the training process is complete.

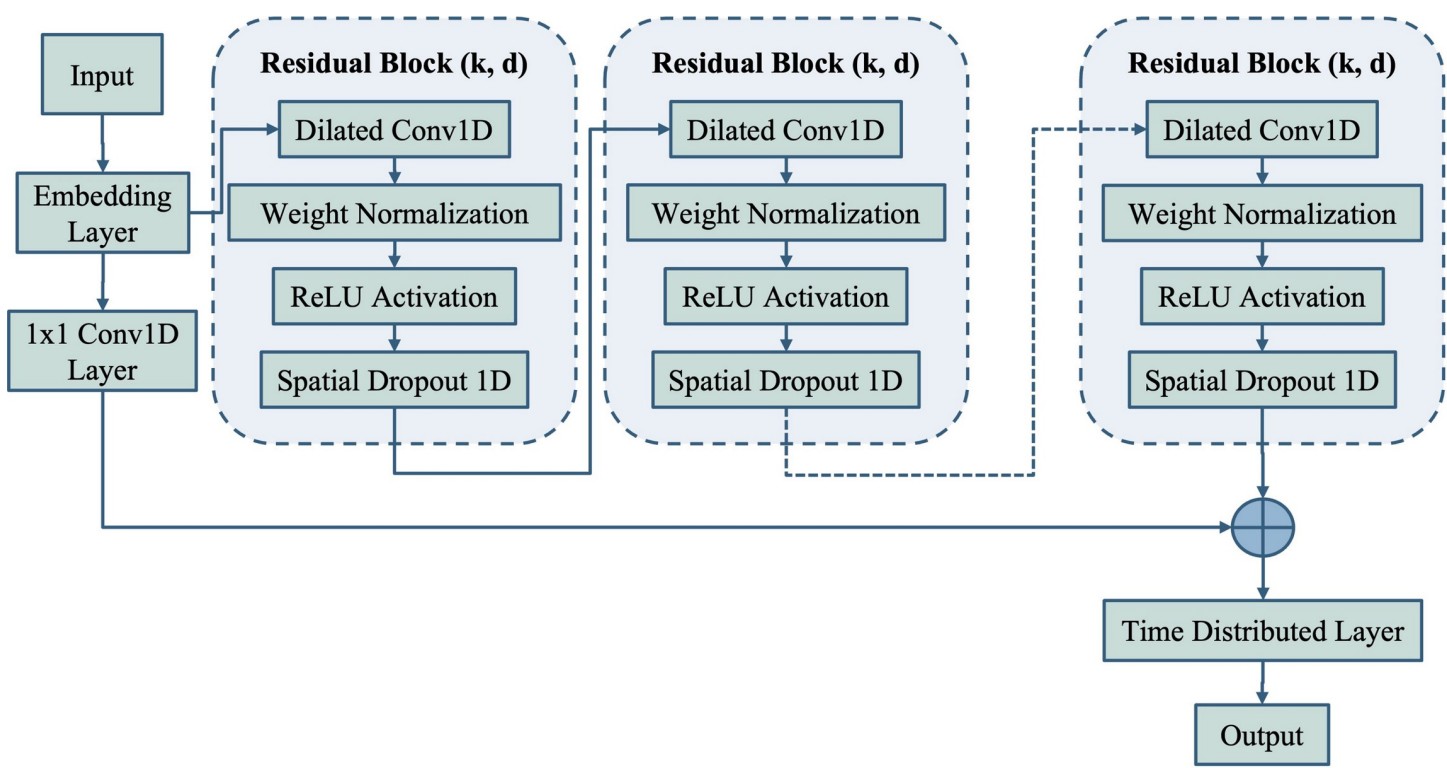

**Fig 6. The proposed TCN model for type and index mutation detection.**

## The proposed TCN model

In this research, a TCN model was proposed to detect type and index mutations in DNA sequence datasets. At the beginning of its development, TCN was widely used in the case of time series data and language models. Additionally, in some sequential modeling tasks, TCN outperform RNN-based methods that were the state of the art prior to its development. TCN has several advantages, including the ability to process sequence inputs in parallel, resulting in faster computations, a flexible receptive field size, the ability to avoid exploding or vanishing gradients, and a shared filter that can be used on different layers, requiring smaller computational memory.

Fig 6 shows the architecture of the proposed TCN model, which consists of embedding layers, residual blocks, 1x1 one dimensional convolution (Conv1D) layers, and time distributed layers. The sequence input that enters the TCN model is the result of DNA sequence mapping using integer, 2-mers, or 3-mers mapping techniques, resulting in a numerical sequence. In the embedding layer section, numerical sequences with dimensions of *1xn* (where n is the length of the sequence) are converted into one-hot representation, resulting in a sequence with dimensions *m+1xn*. Here, m represents the sum of variations in numerical values in all sequences, and the number 1 represents one padding character. The value of m can be determined using Eq 1, where *k* represents the number of nucleotides converted into one value. For example, in the integer mapping technique (1-mers mapping), $k = 1$; in the 2-mers mapping technique, $k = 2$; and in the 3-mers mapping technique, $k = 3$.

$$m = 2^{k+1} \tag{1}$$

The TCN model uses the concept of causal convolution, where in the convolution process only takes into account the element *t* and before it for the calculation, preventing any leakage of information from the element after *t*. The development of the TCN model, compared to basic causal convolution, lies in the residual block that contains a dilated Conv1D and a residual connection. The dilated Conv1D is used to activate an exponentially large receptive field, allowing the model to remember patterns in DNA sequences for longer periods than the standard causal convolution. In the convolution process, the dilated Conv1D uses a dilatation factor (*d*) in addition to the kernel size *k*. The dilatation factor increase exponentially with the depth of the TCN architecture. When the dilatation factor is $d = 1$, the dilated convolution becomes the standard causal convolution. The process of dilated convolution *F* on the index element t of the input sequence *x* with filter *f* is defined with Eq 2.

$$F(x_t) = (x *_d f)(t) = \sum_{i=0}^{k-1} f(i) \cdot x_{t-d \cdot i} \tag{2}$$

Weight normalization is used to solve the problem of exploding gradients by re-parameterizing each parameter after the convolution process [45]. Afterward, ReLU activation is needed to add non-linearity to the built model, preventing it from being a very complex linear model. The ReLU activation function is defined in Eq 3, where *x* is the input value that goes into the activation function, and *max* a function that finds the maximum value among the given values [46]. Furthermore, spatial dropout 1D is a regularization technique used to avoid overfitting, where the trained model becomes too fixated on the patterns in the training data and has a low ability to generalize to new data. Spatial dropout is done by disabling all elements on the feature map, setting the the value to 0 [47]. A residual connection in a residual block is achieved by adding input *x* to the convolutional output of layer *F(x)*, with *o* being the output of residual connection (Eq 4). The residual connection aims to overcome the vanishing gradient problem

that often occurs in many Deep Neural Network methods. Additionally, the 1x1 Conv1D is an identity map used when residual inputs and outputs have different dimensions.

$$Activation(x) = \max(0, x) \tag{3}$$

$$o = Activation(x + F(x)) \tag{4}$$

In this research, a time distributed layer was used as an output layer as the proposed model is a sequential labelling model, where the length of the output sequence is equal to the input sequence. With the proposed model, each nucleotide in the DNA sequence can be classified into a mutated nucleotide label or a normal nucleotide. The system's performance was measured using the precision, recall, and F1-score metrics because the number of mutated nucleotides is much less than the number of normal nucleotides. Precision expresses the ratio of the number of correctly predicted mutations to the number of predicted mutations (Eq 5), while recall represents the ratio of correctly predicted mutations to the actual number of mutations (Eq 6). F1-score is a harmonic mean between precision and recall (Eq 7) [48]. Precision, recall, and F1-score values range from 0 to 1.

$$Precision = TP/(TP + FP) \tag{5}$$

$$Recall = TP/(TP + FN) \tag{6}$$

$$F1 - score = (2 * Precision * Recall)/(Precision + Recall) \tag{7}$$

## Results and discussions

In this section, the experimental results are presented along with the analysis of the type and index mutation detection model on DNA sequences using the TCN model. In the first experiment, the data used consists of 21 genes from 81,272 samples of breast cancer patients and was acquired from the COSMIC database (COSMIC dataset). The dataset is divided into train and test sets in a ratio of 90:10. The parameters observed include DNA mapping techniques, sampling data, number and size kernel, and dilatation factor. The observed DNA mapping techniques include integer mapping, 2-mers, and 3-mers mapping, with or without sampling data (Random Under Side/RUS), the number of kernel (128 and 256), kernel sizes (8, 16, and 32), and dilatation factors ([1, 2, 4, 8, 16, 32] and [1, 2, 4, 8, 16, 32, 64]). The static parameters used include a sub-sequence length of 150 bp with an overlap 50 bp, a learning rate 0.0005, a dropout rate of 0.1, a batch size of 256, and the number of epochs set to 100. The TCN models were trained using Adam's optimization algorithm [49] in python on Google Colaboratory Pro+.

Table 3 presents the results of the proposed model hyperparameters observation on the COSMIC dataset. The table shows that DNA mapping techniques greatly affect the performance of the built model. The 2-mers and 3-mers mapping techniques can improve the F1-score testing by 30%-34% when compared to integer mapping techniques. This is because the proposed TCN model has a receptive field that allows to better remember the patterns contained in the data if the processed input data has more variation. In the integer mapping technique, the variation in input data values is limited to only 4 values: A, C, T, and G, whereas in the 2-mers mapping technique, the variation in input values amounts to 16 values (as shown in Table 1), and in the 3-mers mapping technique, the variation in input values amounts to 64 values (as shown in Table 2).

Table 3. Hyperparameter tuning of TCN model on COSMIC dataset.

| DNA Mapping | Data Sampling | #Kernel | Kernel Size | Dilatation Factor | Validation F-Score | Testing Performance | | |
|---|---|---|---|---|---|---|---|---|
| | | | | | | Precision | Recall | F1-Score |
| Integer | - | 128 | 8 | [1, 2, 4, 8, 16, 32] | 0.6135 | 0.8629 | 0.5553 | 0.5985 |
| Integer | - | 128 | 16 | [1, 2, 4, 8, 16, 32] | 0.6135 | 0.8871 | 0.5627 | 0.6034 |
| Integer | RUS | 128 | 8 | [1, 2, 4, 8, 16, 32, 64] | 0.6462 | 0.5185 | 0.5601 | 0.5377 |
| Integer | RUS | 128 | 16 | [1, 2, 4, 8, 16, 32, 64] | 0.6452 | 0.5203 | 0.5515 | 0.5349 |
| 2-mers | - | 128 | 8 | [1, 2, 4, 8, 16, 32, 64] | 0.9217 | 0.9603 | 0.8517 | 0.9019 |
| 2-mers | - | 128 | 8 | [1, 2, 4, 8, 16, 32] | 0.9298 | 0.9645 | 0.8439 | 0.9016 |
| 2-mers | - | 128 | 16 | [1, 2, 4, 8, 16, 32] | 0.9333 | 0.9609 | 0.8469 | 0.8995 |
| 2-mers | - | 128 | 16 | [1, 2, 4, 8, 16, 32] | 0.8986 | 0.9680 | 0.8609 | **0.9108** |
| 2-mers | - | 128 | 32 | [1, 2, 4, 8, 16, 32] | 0.9464 | 0.9647 | 0.8329 | 0.8930 |
| 2-mers | - | 256 | 16 | [1, 2, 4, 8, 16, 32] | 0.9168 | 0.9666 | 0.8521 | 0.9050 |
| 3-mers | - | 128 | 16 | [1, 2, 4, 8, 16, 32] | 0.9481 | 0.9879 | 0.9050 | **0.9443** |

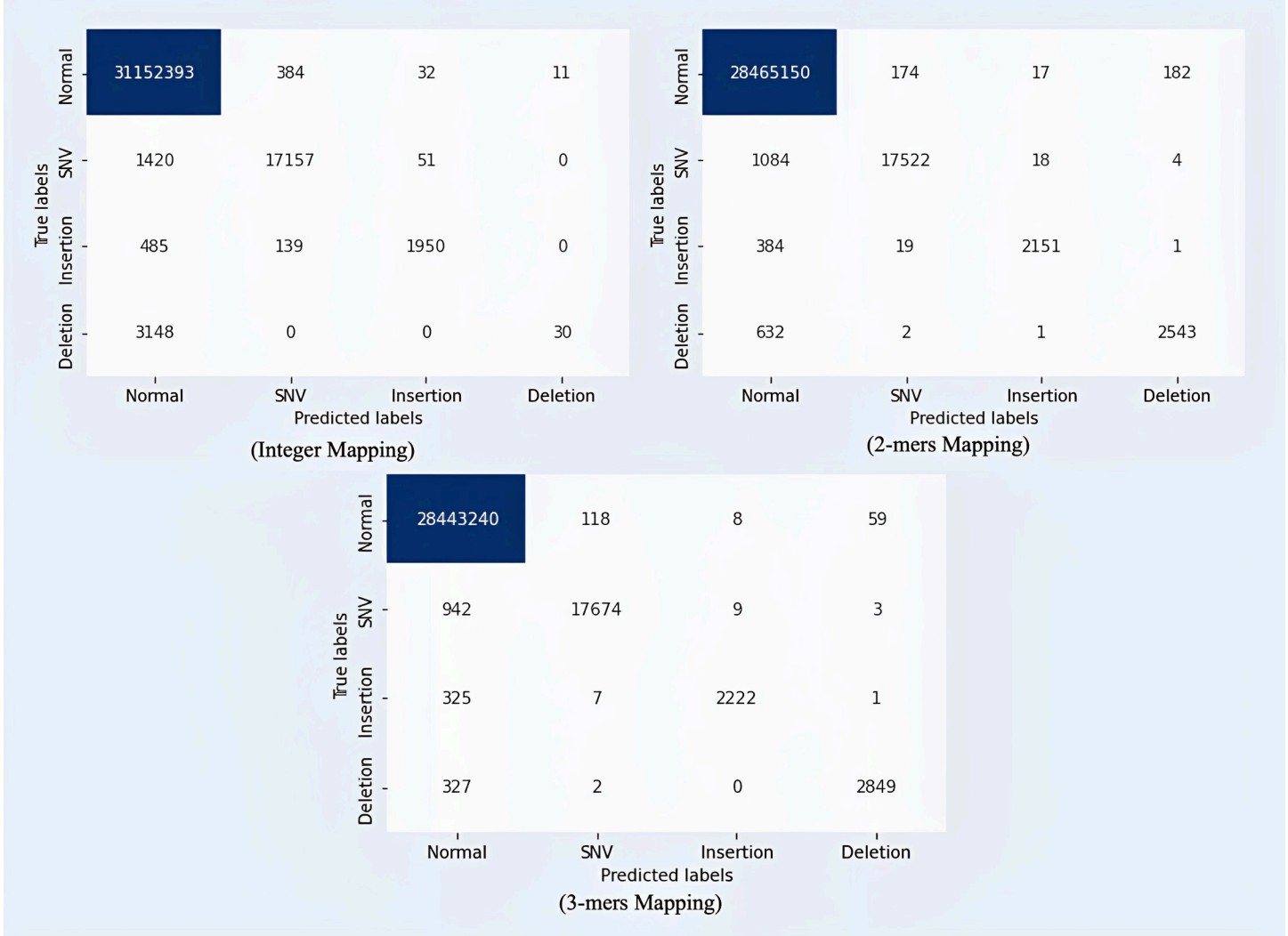

(Integer Mapping)

(2-mers Mapping)

(3-mers Mapping)

**Fig 7. Confusion matrix comparison of the proposed model using integer mapping, 2-mers mapping, and 3-mers mapping.**

This can also be observed in the confusion matrix shown in Fig 7, where the majority of nucleotides with deletion mutations are predicted to be normal nucleotides. In other words, the DNA sequence pattern represented using the integer mapping technique does not provide enough information for the TCN model. After applying the 2-mers mapping technique and the 3-mers mapping technique, many errors in detecting deletion mutations have been corrected, and improvements have also been made in the detection of insertion and SNV mutations. The ROC curve in Fig 8 also demonstrates the ability of the proposed TCN model to detect type and index mutation. When using the integer mapping technique, the AUC value for deletion mutations is only 0.5, indicating that the model cannot distinguish deletion mutations from other types of mutations or from normal nucleotides. However, when using the 2-mers and 3-mers techniques, the model was able to effectively distinguish deletion mutations from other labels. The AUC values achieved by the proposed TCN model using the 2-mers mapping technique are 0.97 for SNV mutations, 0.92 for insertion mutations, and 0.9 for deletion mutations. And when using the 3-mers mapping technique, the proposed TCN model can achieve an AUC value of 0.97 for SNV mutations, 0.93 for insertion mutations, and 0.95 for deletion mutations.

For data sampling technique that is implemented to balance the amount of data in different classes in the training data, it only has a negative influence of 6% on F1-score testing. Therefore, the use of sampling data will be adjusted based on the amount of data and RAM available during the training process. Regarding the number of kernels, kernel sizes, and dilation factors, increasing their values will add complexity to the built model, but it does not guarantee an improvement in model performance in the event of overfitting. Hence, it is crucial to find optimal hyperparameters for specific problems and data types. In the testing using COSMIC dataset, the best testing performance was achieved with 2-mers mapping technique, which yielded F1-score of 0.9056. The 3-mers mapping technique resulted in the highest F1-score of 0.9427, with 128 kernels, 16 kernel sizes, and dilatation factors of [1, 2, 4, 8, 16, 32].

After obtaining the optimal hyperparameters for the proposed TCN model to detect the type and index mutations in the COSMIC dataset, which was acquired from the COSMIC dataset, the next step was to test the model's ability to detect SNV index mutation on the RSCM dataset consisting of three datasets: 100k, 250k, and 500k datasets. The hyperparameter used in data preprocessing included 2-mers and 3-mers DNA mapping techniques, as well as RUS data sampling techniques due to RAM limitations. The hyperparameters for the TCN model included number of kernels 128, a kernel size of 16, a dilation factor of [1, 2, 4, 8, 16, 32], a learning rate of 0.0005, and dropout rate of 0.1. The proposed model was also compared with the BiLSTM model that was previously used in the study [21]. The best hyperparameters for the BiLSTM model obtained in the study included integer mapping, a 2-layer BiLSTM, 256 LSTM units, a dropout rate of 0.2, and a learning rate of 0.0001. Both models were trained using Adam's optimization algorithm [49] with a batch size of 256 and 100 epochs.

Fig 9 shows a performance comparison of the proposed TCN model on COSMIC dataset and RSCM dataset, which consist of 100k, 250k, and 500k datasets, as well as a comparison with the BiLSTM model. The BiLSTM model perform slightly better than the proposed TCN model in in detecting index mutation on the COSMIC dataset, with an F1-score difference of 0.0115. However, the proposed TCN model is superior in detecting index mutation on the RSCM dataset in all three datasets (100k, 250k, and 500k). This proves that the proposed TCN model is more robust in detecting data with high heterogeneity, where in the RSCM dataset, sequence variations are spread across 24 different chromosomes (as shown in Fig 5), while in the COSMIC dataset, the sequence variation is only spread across 21 genes. Furthermore, the proposed TCN model achieves a higher F1-score with the use of more sequences, as evidenced by the increase in F1-score on the 100k, 250k, and 500k datasets. Additionally, as shown in Fig

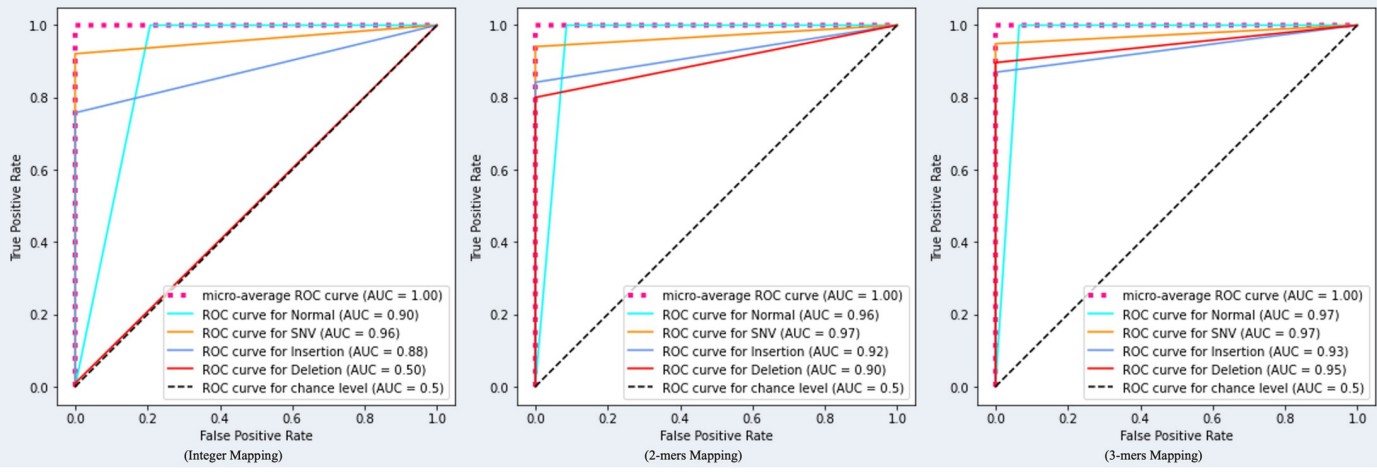

**Fig 8. ROC curve comparison of the proposed model using integer mapping, 2-mers mapping, and 3-mers mapping.**

10, the proposed TCN model processes one sequence in an average of 0.0034 seconds, while the BiLSTM model processes one sequence is an average of 0.222 seconds. Therefore, the proposed TCN model can detect index mutation six times faster than the BiLSTM model. The BiLSTM model requires a larger number of units (256 units) to achieve high performance, whereas the proposed TCN model only requires 128 kernels, making it faster.

The time complexity of the proposed TCN model is influenced by several factors, including the number of layers ($L$), sequence length ($n$), kernel size ($K$), and dilation factor ($d$). A convolution layer with a kernel size $K$ has a time complexity of O($nK$), while the time complexity of a residual block with a dilation factor of $d$ is O($n(K+d)$) time complexity. Thus, the time

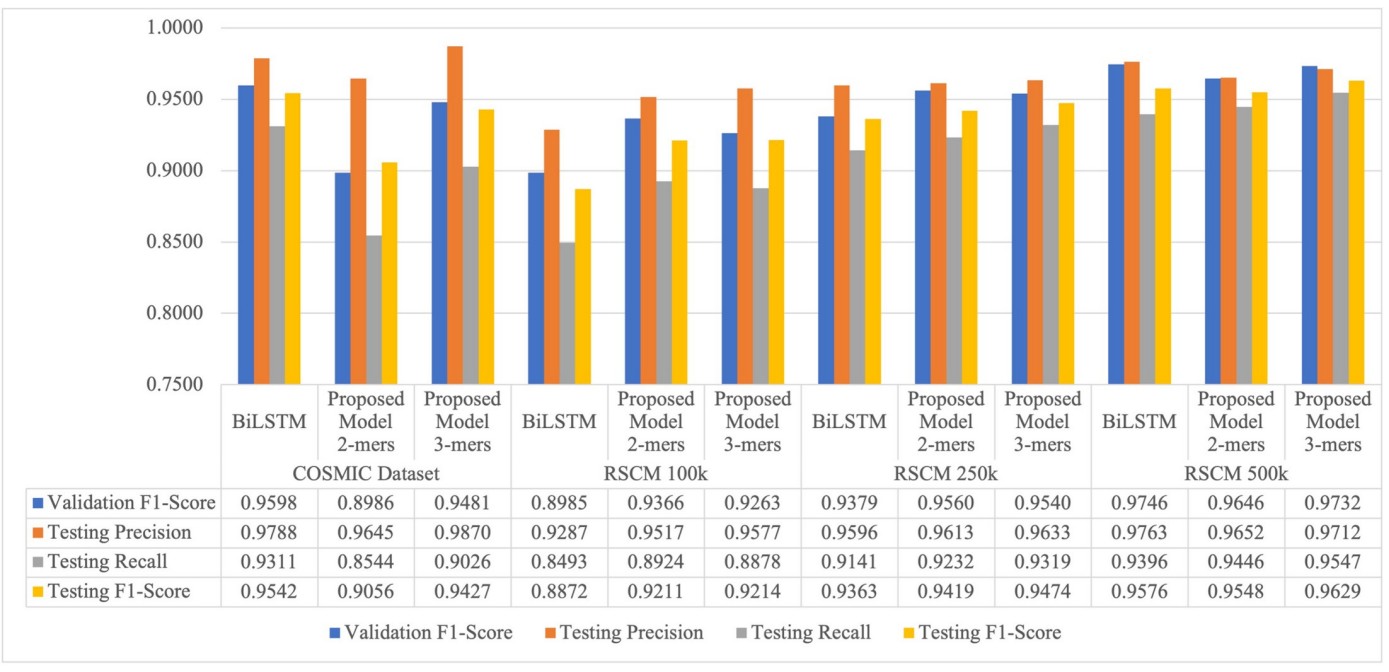

**Fig 9. Performance comparison on COSMIC dataset and RSCM dataset.**

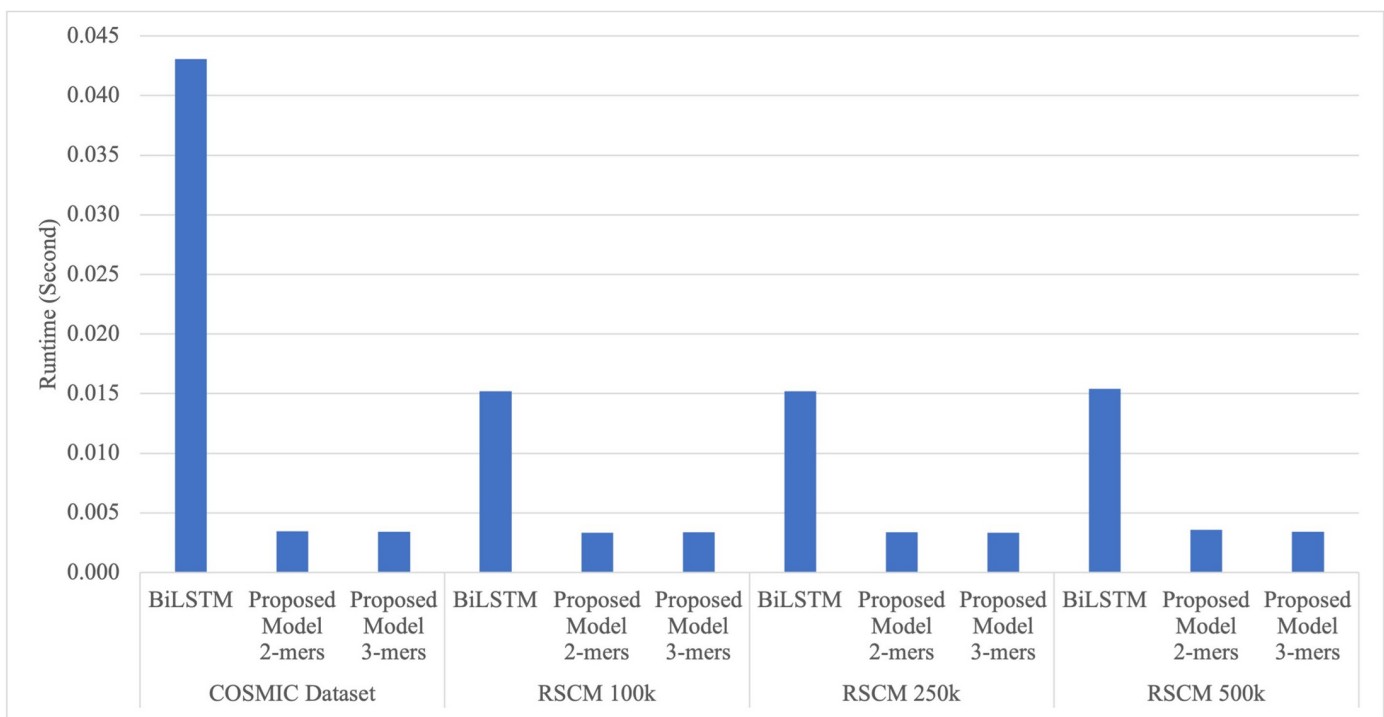

**Fig 10. Runtime comparison to detect index mutation on one sequence.**

complexity of the TCN model becomes $O(nL(K+d))$. The model's complexity is also influenced by the number of sequences in the training data and the number of epochs used. At the testing stage, the time complexity is only calculated for one test sequence because a pre-built model can be used to detect the type and index mutation without aligning it to reference sequences. Additionally, the proposed TCN model can process calculations in the kernel in parallel. The proposed TCN model achieved the highest F1-scores of 0.9443 for COSMIC dataset, 0.9214 for the RSCM 100k dataset, 0.9474 for the RSCM 250k dataset, and 0.9629 for the RSCM 500k dataset.

Finally, type and index mutation detection in this research can be used for early diagnosis of cancer or other diseases (with DNA sequence data related to the disease), allowing treatment to be carried out as early as possible. In cancer, mutation detection can also be used for targeted treatments, which are more effective than conventional chemotherapy. Mutation detection can also be used for personalized medicine, which can be adapted to the patient's genetic profile. Additionally, for healthy individuals, mutation detection can be used to determine their risk of developing certain diseases, so that they can take preventive measures and make lifestyle changes to minimize the likelihood of contracting the disease.

The proposed TCN model has several advantages over other related work, but it still has some limitations that can be addressed in future research. Firstly, the dataset used in this research is limited to breast cancer datasets acquired from the COSMIC database and RSCM. Therefore, it would be beneficial to add more dataset variations in future research, and it would be helpful if the human DNA dataset could be easily obtained with anonymous patient identity. Secondly, the proposed TCN model can also be applied to predict mutations in other genetic diseases, such as diabetes, Alzheimer's, thalassemia, color blindness, and other diseases.

## Conclusion

In this paper, a type and index mutation detection model was proposed using the TCN model and 2-mers and 3-mers mapping techniques on DNA sequence data from breast cancer patients. The architecture of the proposed TCN model is specifically designed for sequential labeling tasks on DNA sequences data, allowing for the detection of mutation types for each nucleotide in the sequence, resulting in obtaining index mutations in the sequence. The proposed model can also detect one or more mutations in one input sequence, along with the index of where the mutation occurs, with a faster detection time. The data used in this research includes the COSMIC dataset, which comprises of a combination of 21 genes associated with breast cancer and was acquired from the COSMIC database, and the RSCM dataset, which consists of 100k, 250k, and 500k datasets acquired from breast cancer patients at Cipto Mangunkusumo National Hospital, Jakarta, Indonesia. The 2-mers and 3-mers mapping techniques were observed to increase F1-score testing by 30–34% compared to the integer mapping technique in the COSMIC dataset. In testing using the RSCM dataset, the proposed model's performance was compared with the BiLSTM model. Although BiLSTM performed slightly better than the proposed TCN model on the COSMIC dataset with an F1-score difference of 0.0115, BiLSTM requires 256 LSTM units, resulting in longer processing time. The proposed TCN model is superior in detecting index mutation in RSCM datasets, which exhibit a higher degree of heterogeneity, and can detect index mutation six times faster than BiLSTM models. The proposed TCN model achieved the highest F1-scores of 0.9427 for COSMIC dataset and 0.9629 for RSCM dataset.

## Author Contributions

**Conceptualization:** Untari Novia Wisesty, Tati Rajab Mengko, Ayu Purwarianti, Adi Pancoro.

**Data curation:** Untari Novia Wisesty.

**Methodology:** Untari Novia Wisesty, Ayu Purwarianti.

**Software:** Untari Novia Wisesty.

**Supervision:** Tati Rajab Mengko, Ayu Purwarianti, Adi Pancoro.

**Validation:** Tati Rajab Mengko, Ayu Purwarianti, Adi Pancoro.

**Writing – original draft:** Untari Novia Wisesty.

**Writing – review & editing:** Tati Rajab Mengko, Ayu Purwarianti, Adi Pancoro.

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
