## [Decision Letter · Decision Letter 0]

24 Feb 2023

PONE-D-22-32883Temporal Convolutional Network for a Fast DNA Mutation Detection in Breast Cancer DataPLOS ONE

Dear Dr. Mengko,

Thank you for submitting your manuscript to PLOS ONE. After careful consideration, we feel that it has merit but does not fully meet PLOS ONE’s publication criteria as it currently stands. Therefore, we invite you to submit a revised version of the manuscript that addresses the points raised during the review process.

We look forward to receiving your revised manuscript.

Kind regards,

Mohamed Hammad, Ph.D.

Academic Editor

PLOS ONE

Journal Requirements:

Additional Editor Comments (if provided):

When updating your manuscript, you should elaborate on your points and clarify with references, examples, data, etc. Also, note that if a reviewer suggested references, you should only add those that are relevant to your work if you feel they strengthen your article. Recommending references to specific publications is not appropriate for reviewers.

Reviewers' comments:

Reviewer's Responses to Questions

**Comments to the Author**

1. Is the manuscript technically sound, and do the data support the conclusions?

Reviewer #1: Yes

Reviewer #2: Partly

2. Has the statistical analysis been performed appropriately and rigorously? 

Reviewer #1: No

Reviewer #2: Yes

3. Have the authors made all data underlying the findings in their manuscript fully available?

Reviewer #1: Yes

Reviewer #2: Yes

4. Is the manuscript presented in an intelligible fashion and written in standard English?

Reviewer #1: Yes

Reviewer #2: No

5. Review Comments to the Author

Reviewer #1: The paper titled “Temporal Convolutional Network for a Fast DNA Mutation Detection in Breast Cancer Data” developed a temporal CNN model. I have some comments to improve the readability and clarity of the manuscript.

1. Abstract of the paper does not highlight the research findings. Abstract section may be rewritten to make more meaningful.

2. Introduction section can be extended to add the issues in the context of the existing work and how the proposed algorithms/approach can be used to overcome this.

3. In the Introduction section, the proposed method's new features and the results' main advantages over others should be clearly described.

4. The problems of this work are not clearly stated. There is ambiguity in the statement understanding.

5. More clarifications and highlights about the research gaps in the related works section needs to be included.

6. If no one has proposed a method like the proposed algorithm before, this claim should be highlighted much more.

7. A comparison with the state-of-the-art in the form of a table should be added.

8. The authors provided details about the implementation setup and working environment. However, some training information should also be given in the experimental section.

9. Discuss research contributions.

Indicate practical advantages (in at least one separate paragraph),

discuss research limitations (at least one separate paragraph), and

supply 2-3 solid and insightful future research suggestions.

10. Authors need to provide the merits of this study vs other review studies.

11. Authors should add computational complexity.

12. Limitations and the future scope should be added.

13. Qualities of images are poor.

14. Equations are not well explained.

15. The developed network is over-fitted as validation accuracy crossed training accuracy.

16. Authors should conduct re-experimentation by tuning the hyper-parameters.

17. ROC curve is not shown.

18. Conclusions are required re-write with indication of potential results.

19. Please cite the below latest published scripts in your article

@article{sahoo2022real,

title={Real-Time Hand Gesture Recognition Using Fine-Tuned Convolutional Neural Network},

author={Sahoo, Jaya Prakash and Prakash, Allam Jaya and P{\\l}awiak, Pawe{\\l} and Samantray, Saunak},

journal={Sensors},

volume={22},

number={3},

pages={706},

year={2022},

publisher={MDPI}

}

@incollection{locharla2022eeg,

title={EEG-based deep learning neural net for apnea detection},

author={Locharla, Govinda Rao and Pogiri, Revathi and Allam, Jaya Prakash},

booktitle={Artificial Intelligence-Based Brain-Computer Interface},

pages={203--215},

year={2022},

publisher={Elsevier}

}

@incollection{prakash2021automated,

title={An Automated Diagnosis System for Cardiac Arrhythmia Classification},

author={Prakash, Allam Jaya and Samantray, Saunak and Bala, CH Laxmi and Narayana, YV},

booktitle={Analysis of Medical Modalities for Improved Diagnosis in Modern Healthcare},

pages={301--313},

year={2021},

publisher={CRC Press}

}

@incollection{allam2022customized,

title={Customized deep learning algorithm for drowsiness detection using single-channel EEG signal},

author={Allam, Jaya Prakash and Samantray, Saunak and Behara, Chinmaya and Kurkute, Ketan Kishor and Sinha, Vikas Kumar},

booktitle={Artificial Intelligence-Based Brain-Computer Interface},

pages={189--201},

year={2022},

publisher={Elsevier}

}

@article{patro2022efficient,

title={An efficient optimized feature selection with machine learning approach for ECG biometric recognition},

author={Patro, Kiran Kumar and Jaya Prakash, Allam and Jayamanmadha Rao, M and Rajesh Kumar, P},

journal={IETE Journal of Research},

volume={68},

number={4},

pages={2743--2754},

year={2022},

publisher={Taylor \\& Francis}

}

@article{sinha2021smartphone,

title={Smartphone-Based Human Sitting Behaviors Recognition Using Inertial Sensor},

author={Sinha, Vikas Kumar and Patro, Kiran Kumar and P{\\l}awiak, Pawe{\\l} and Prakash, Allam Jaya},

journal={Sensors},

volume={21},

number={19},

pages={6652},

year={2021},

publisher={MDPI}

}

@article{prakash2022capsule,

title={Capsule Network for the Identification of Individuals Using Quantized ECG Signal Images},

author={Prakash, Allam Jaya},

journal={IEEE Sensors Letters},

volume={6},

number={8},

pages={1--4},

year={2022},

publisher={IEEE}

}

@article{hammad2022automated,

title={Automated detection of myocardial infarction and heart conduction disorders based on feature selection and a deep learning model},

author={Hammad, Mohamed and Chelloug, Samia Allaoua and Alkanhel, Reem and Prakash, Allam Jaya and Muthanna, Ammar and Elgendy, Ibrahim A and P{\\l}awiak, Pawe{\\l}},

journal={Sensors},

volume={22},

number={17},

pages={6503},

year={2022},

publisher={MDPI}

}

@article{kiran2022hybrid,

title={A hybrid approach of a deep learning technique for real-time ECG beat detection},

author={Kiran Kumar Patro, Allam Jaya Prakash, Saunak Samantray, Joanna P{\\l}awiak, Ryszard Tadeusiewicz, Pawe{\\l} P{\\l}awiak},

journal={Int. J. Appl. Math. Comput. Sci.},

volume={32},

number={3},

pages={455--465},

year={2022},

publisher={{\\copyright} AMCS 2022 D\\&C Marcin Paszek}

}

@article{prakash2022baed,

title={BAED: A secured biometric authentication system using ECG signal based on deep learning techniques},

author={Prakash, Allam Jaya and Patro, Kiran Kumar and Hammad, Mohamed and Tadeusiewicz, Ryszard and P{\\l}awiak, Pawe{\\l}},

journal={Biocybernetics and Biomedical Engineering},

volume={42},

number={4},

pages={1081--1093},

year={2022},

publisher={Elsevier}

}

@article{pedada2023novel,

title={A novel approach for brain tumour detection using deep learning based technique},

author={Pedada, Kameswara Rao and Rao, Bhujanga and Patro, Kiran Kumar and Allam, Jaya Prakash and Jamjoom, Mona M and Samee, Nagwan Abdel},

journal={Biomedical Signal Processing and Control},

volume={82},

pages={104549},

year={2023},

publisher={Elsevier}

}

Reviewer #2: This paper presents early detection of breast cancer by DNA sequences using temporal convolutional network. The basic idea sounds good but needs some recommendations to improve the quality of the manuscript.

1. Re-write the abstract by significance of your work, limitations of previous works and how your work overcomes those? (state briefly).

2. Novelty of the contributions should be highlighted.

3. Better to include a table and summarize previous works along with limitations in the introduction section.

4. Refer the following important works on importance of CNN

https://doi.org/10.1016/j.bbe.2022.08.004

https://doi.org/10.3390/ijms24010146

http://dx.doi.org/10.34768/amcs-2022-0033

https://doi.org/10.1016/j.acra.2022.12.038

5. Spell out each acronym the first time used in the body of the paper.

6. Significance of the figure 2 needs to be included.

7. What is the purpose of Time distributed layer, needs to be addressed.

8. Include specific author contributions at the end of the introduction section.

9. Cite the datasets in the reference section. don’t cite like Dataset21

10. Cite all the equations, figures and tables in the manuscript.

11. Include software and Hardware requirements in the results section.

12. In performance parameters calculation why author not much concentrated on accuracy?

13. A Comparison Table required to compare recent similar works.

14. Improve the quality of the all figures.

15. Read and recheck whole paper and correct grammatical and over spacing errors.

Overall, the idea was good, but needs improvement in the presentation of the work.

6. PLOS authors have the option to publish the peer review history of their article (what does this mean?). If published, this will include your full peer review and any attached files.

Reviewer #1: No

Reviewer #2: **Yes: **Dr Kiran Kumar Patro

---

## [Author Response · Author response to Decision Letter 0]

28 Mar 2023

Reviewer #1: The paper titled “Temporal Convolutional Network for a Fast DNA Mutation Detection in Breast Cancer Data” developed a temporal CNN model. I have some comments to improve the readability and clarity of the manuscript.

Concern #1:

Abstract of the paper does not highlight the research findings. Abstract section may be rewritten to make more meaningful.

Author response:

The abstract has been rewritten:

Early detection of breast cancer can be achieved through mutation detection in DNA sequences, which can be acquired through patient blood samples. Mutation detection can be performed using alignment and machine learning techniques. However, alignment techniques require reference sequences, and machine learning techniques still cannot predict index mutation and require supporting tools. Therefore, in this research, a Temporal Convolutional Network (TCN) model was proposed to detect the type and index mutation faster and without reference sequences and supporting tools. The architecture of the proposed TCN model is specifically designed for sequential labeling tasks on DNA sequence data. This allows for the detection of the mutation type of each nucleotide in the sequence, and if the nucleotide has a mutation, the index mutation can be obtained. The proposed model also uses 2-mers and 3-mers mapping techniques to improve detection performance. Based on the tests that have been carried out, the proposed TCN model can achieve the highest F1-score of 0.9443 for COSMIC dataset and 0.9629 for RSCM dataset, Additionally, the proposed TCN model can detect index mutation six times faster than BiLSTM model. Furthermore, the proposed model can detect type and index mutations based on the patient's DNA sequence, without the need for reference sequences or other additional tools.

Concern #2:

Introduction section can be extended to add the issues in the context of the existing work and how the proposed algorithms/approach can be used to overcome this.

Author response:

We add 3 paragraph in Introduction section (paragraph 3, 4, and 5) about the issues in existing work.

Concern #3:

In the Introduction section, the proposed method's new features and the results' main advantages over others should be clearly described.

Author response:

We describe the novelty of the proposed model in the last paragraph of Introduction section:

The architecture of the proposed TCN model is specifically designed for sequential labeling tasks on DNA sequence data. This allows for the detection of the mutation type of each nucleotide in the sequence, and if the nucleotide has a mutation, the index mutation can be obtained. The proposed model can also detect one or more mutations in one input sequence along with the index where the mutation occurs with a faster detection time. 2-mers and 3-mers mapping techniques are also proposed in this research to improve the performance level in detecting type and index mutation. The proposed TCN model succeeded in achieving an F1-score of 0.9427 for the COSMIC dataset and 0.9629 for the RSCM dataset, and was six times faster than BiLSTM model. Fig. 1 shows the advantages and research gap between the proposed TCN model and the related work. Finnaly, in this research, all authors planned and conceptualized the conducted research, U.W. carried out the implementation and experimentation of the proposed model, analysis the results, and wrote the draft paper, T.M. led and supervised the project, while A.P. and A.P. analyzed the results and proofread the manuscript.

Concern #4:

The problems of this work are not clearly stated. There is ambiguity in the statement understanding.

Author response:

We add 3 paragraph in Introduction section (paragraph 3, 4, and 5) about the issues in existing work.

In the field of bioinformatics, two approaches are generally used to detect DNA mutations, namely the alignment approach and machine learning approaches. Several studies have been developed using the alignment approach, such as AutoCSA (8), IsoMut (9), sweep-based seed processing (10), MutClustSW (11), and Bivartect (12). The main problems that occur when using the alignment approach are the requirement of reference sequences and the need for a long test time to detect mutations in new sequences because the new sequences must be matched with all relevant reference sequences.

The second approach in detecting mutations using DNA sequence data is the machine learning approach. This approach has the advantage that the time required to test new samples is relatively faster than the alignment approach, because the new samples to be tested are entered into the model without being matched one by one to the relevant reference samples. Several studies have used the machine learning approach, namely SNVHMM (13), Bayesian based Somatic Structural Variation (14), Functional Link Artificial Neural Network (15), Tumor Covariate Signature Model (16), DriverML (17), Multi-Modal Deep Boltzmann Machine (18), NeoSomatic (19), circulating tumor DNA detection (20).

The machine learning approach has several drawbacks, including the lower level of accuracy in mutation detection when compared to alignment techniques. Additionally, most existing research only classifies whether a sequence has mutations or not, without predicting where the mutation occurred. In our previous research, we used the Bidirectional Long Short-Term Memory (BiLSTM) model to detect mutations in DNA sequences (21). However, the BiLSTM still requires some time to detect mutations, and the data used is limited to single gene datasets in lung cancer acquired from the Catalog of Somatic Mutations in Cancer (COSMIC database) (22).

Concern #5:

More clarifications and highlights about the research gaps in the related works section needs to be included.

Author response:

We add Fig 1 to highlights the research gaps between the related works and proposed TCN model in Introduction section.

Concern #6:

If no one has proposed a method like the proposed algorithm before, this claim should be highlighted much more.

Author response:

We add some sentence below to highlights our proposed model:

In our previous research, we used the Bidirectional Long Short-Term Memory (BiLSTM) model to detect mutations in DNA sequences (21). However, the BiLSTM still requires some time to detect mutations, and the data used is limited to single gene datasets in lung cancer acquired from the Catalog of Somatic Mutations in Cancer (COSMIC database) (22). � Paragraph 5 in Introduction Section.

The proposed TCN model succeeded in achieving an F1-score of 0.9427 for the COSMIC dataset and 0.9629 for the RSCM dataset, and was six times faster than BiLSTM model. � Last paragraph in Introduction section.

Concern #7:

A comparison with the state-of-the-art in the form of a table should be added.

Author response:

We add Fig 1 to highlights the comparison between the related works and proposed TCN model in Introduction section.

Concern #8:

The authors provided details about the implementation setup and working environment. However, some training information should also be given in the experimental section.

Author response:

We refine the sentences in first paragraph in Result and Discussion section about the training information:

In this section, the experimental results are presented along with the analysis of the type and index mutation detection model on DNA sequences using the TCN model. In the first experiment, the data used consists of 21 genes from 81,272 samples of breast cancer patients and was acquired from the COSMIC database (COSMIC dataset). The dataset is divided into train and test sets in a ratio of 90:10. The parameters observed include DNA mapping techniques, sampling data, number and size kernel, and dilatation factor. The observed DNA mapping techniques include integer mapping, 2-mers, and 3-mers mapping, with or without sampling data (Random Under Side/RUS), the number of kernel (128 and 256), kernel sizes (8, 16, and 32), and dilatation factors ([1, 2, 4, 8, 16, 32] and [1, 2, 4, 8, 16, 32, 64]). The static parameters used include a sub-sequence length of 150 bp with an overlap 50 bp, a learning rate 0.0005, a dropout rate of 0.1, a batch size of 256, and the number of epochs set to 100. The TCN models were trained using Adam's optimization algorithm (49) in python on Google Colaboratory Pro+.

Concern #9:

Discuss research contributions.

• Indicate practical advantages (in at least one separate paragraph),

• discuss research limitations (at least one separate paragraph), and

• supply 2-3 solid and insightful future research suggestions.

Author response:

We add 2 last paragraphs in Results and Discussion section:

Finally, type and index mutation detection in this research can be used for early diagnosis of cancer or other diseases (with DNA sequence data related to the disease), allowing treatment to be carried out as early as possible. In cancer, mutation detection can also be used for targeted treatments, which are more effective than conventional chemotherapy. Mutation detection can also be used for personalized medicine, which can be adapted to the patient's genetic profile. Additionally, for healthy individuals, mutation detection can be used to determine their risk of developing certain diseases, so that they can take preventive measures and make lifestyle changes to minimize the likelihood of contracting the disease.

The proposed TCN model has several advantages over other related work, but it still has some limitations that can be addressed in future research. Firstly, the dataset used in this research is limited to breast cancer datasets acquired from the COSMIC database and RSCM. Therefore, it would be beneficial to add more dataset variations in future research, and it would be helpful if the human DNA dataset could be easily obtained with anonymous patient identity. Secondly, the proposed TCN model can also be applied to predict mutations in other genetic diseases, such as diabetes, Alzheimer's, thalassemia, color blindness, and other diseases.

Concern #10:

Authors need to provide the merits of this study vs other review studies.

Author response:

Here are some superiority that differ the proposed model with other studies (last paragraph of Introduction section):

the detection of type and index mutation in DNA sequence data for breast cancer patients using the TCN model. The data inputted is only in the form of DNA sequences and without using reference sequences or other additional tools.

Additionally, this research, also use DNA sequence data from breast cancer patients at Cipto Mangunkusumo National Hospital, Jakarta, Indonesia (RSCM dataset) (40). The data is in the form of sequencing results from patient blood samples in the form of short sequences and have a high degree of heterogeneity and with a larger number of sequences.

The architecture of the proposed TCN model is specifically designed for sequential labeling tasks on DNA sequence data. This allows for the detection of the mutation type of each nucleotide in the sequence, and if the nucleotide has a mutation, the index mutation can be obtained. The proposed model can also detect one or more mutations in one input sequence along with the index where the mutation occurs with a faster detection time.

2-mers and 3-mers mapping techniques are also proposed in this research to improve the performance level in detecting type and index mutation.

We also add Fig 1 to show the advantages of the proposed TCN model compared to the related work.

Concern #11:

Authors should add computational complexity.

Author response:

We add some sentences below in Results and Discussion section:

The time complexity of the proposed TCN model is influenced by several factors, including the number of layers (L), sequence length (n), kernel size (K), and dilation factor (d). A convolution layer with a kernel size K has a time complexity of O(nK), while the time complexity of a residual block with a dilation factor of d is O(n(K+d)) time complexity. Thus, the time complexity of the TCN model becomes O(nL(K+d)). The model's complexity is also influenced by the number of sequences in the training data and the number of epochs used. At the testing stage, the time complexity is only calculated for one test sequence because a pre-built model can be used to detect the type and index mutation without aligning it to reference sequences.

Concern #12:

Limitations and the future scope should be added.

Author response:

We add the limitations and the future research in the last paragraph of Results and Discussions section:

The proposed TCN model has several advantages over other related work, but it still has some limitations that can be addressed in future research. Firstly, the dataset used in this research is limited to breast cancer datasets acquired from the COSMIC database and RSCM. Therefore, it would be beneficial to add more dataset variations in future research, and it would be helpful if the human DNA dataset could be easily obtained with anonymous patient identity. Secondly, the proposed TCN model can also be applied to predict mutations in other genetic diseases, such as diabetes, Alzheimer's, thalassemia, color blindness, and other diseases.

Concern #13:

Qualities of images are poor.

Author response:

We check and refine the low quality of images in our manuscript.

Concern #14:

Equations are not well explained.

Author response:

We add some description of variable in equation 2, 3, and 4. 

Concern #15:

The developed network is over-fitted as validation accuracy crossed training accuracy.

Author response:

We use validation and testing F1-score, where the validation F1-score is calculated based on validation data that is separate from the training data, as well as testing data which is also different from the training and validation data. In Table 3, the validation F1-score is only slightly higher than the F1-score testing value, namely the difference is below 0.12 when using the integer mapping technique and below 0.054 when using the 2-mers mapping technique, so that the TCN model built has not experienced overfitting.

Concern #16:

Authors should conduct re-experimentation by tuning the hyper-parameters.

Author response:

We added and re-tuned the hyperparameters in Table 3 which include DNA mapping techniques, data sampling, number of kernels, kernel size, and dilation factors.

Concern #17:

ROC curve is not shown.

Author response:

We add the ROC curve in Fig. 8, along with the explanation in the Results and Discussion section:

The ROC curve in Fig 8 also demonstrates the ability of the proposed TCN model to detect type and index mutation. When using the integer mapping technique, the AUC value for deletion mutations is only 0.5, indicating that the model cannot distinguish deletion mutations from other types of mutations or from normal nucleotides. However, when using the 2-mers and 3-mers techniques, the model was able to effectively distinguish deletion mutations from other labels. The AUC values achieved by the proposed TCN model using the 2-mers mapping technique are 0.97 for SNV mutations, 0.92 for insertion mutations, and 0.9 for deletion mutations. And when using the 3-mers mapping technique, the proposed TCN model can achieve an AUC value of 0.97 for SNV mutations, 0.93 for insertion mutations, and 0.95 for deletion mutations.

Concern #18:

Conclusions are required re-write with indication of potential results.

Author response:

The conclusion section has been rewritten:

In this paper, a type and index mutation detection model was proposed using the TCN model and 2-mers and 3-mers mapping techniques on DNA sequence data from breast cancer patients. The architecture of the proposed TCN model is specifically designed for sequential labeling tasks on DNA sequences data, allowing for the detection of mutation types for each nucleotide in the sequence, resulting in obtaining index mutations in the sequence. The proposed model can also detect one or more mutations in one input sequence, along with the index of where the mutation occurs, with a faster detection time. The data used in this research includes the COSMIC dataset, which comprises of a combination of 21 genes associated with breast cancer and was acquired from the COSMIC database, and the RSCM dataset, which consists of 100k, 250k, and 500k datasets acquired from breast cancer patients at Cipto Mangunkusumo National Hospital, Jakarta, Indonesia. The 2-mers and 3-mers mapping techniques were observed to increase F1-score testing by 30-34% compared to the integer mapping technique in the COSMIC dataset. In testing using the RSCM dataset, the proposed model's performance was compared with the BiLSTM model. Although BiLSTM performed slightly better than the proposed TCN model on the COSMIC dataset with an F1-score difference of 0.0115, BiLSTM requires 256 LSTM units, resulting in longer processing time. The proposed TCN model is superior in detecting index mutation in RSCM datasets, which exhibit a higher degree of heterogeneity, and can detect index mutation six times faster than BiLSTM models. The proposed TCN model achieved the highest F1-scores of 0.9427 for COSMIC dataset and 0.9629 for RSCM dataset.

Concern #19:

Please cite the below latest published scripts in your article:

title={Real-Time Hand Gesture Recognition Using Fine-Tuned Convolutional Neural Network},

author={Sahoo, Jaya Prakash and Prakash, Allam Jaya and P{\\l}awiak, Pawe{\\l} and Samantray, Saunak}

title={EEG-based deep learning neural net for apnea detection},

author={Locharla, Govinda Rao and Pogiri, Revathi and Allam, Jaya Prakash}

title={An Automated Diagnosis System for Cardiac Arrhythmia Classification},

author={Prakash, Allam Jaya and Samantray, Saunak and Bala, CH Laxmi and Narayana, YV},

title={Customized deep learning algorithm for drowsiness detection using single-channel EEG signal},

author={Allam, Jaya Prakash and Samantray, Saunak and Behara, Chinmaya and Kurkute, Ketan Kishor and Sinha, Vikas Kumar},

title={An efficient optimized feature selection with machine learning approach for ECG biometric recognition},

author={Patro, Kiran Kumar and Jaya Prakash, Allam and Jayamanmadha Rao, M and Rajesh Kumar, P}

title={Smartphone-Based Human Sitting Behaviors Recognition Using Inertial Sensor},

author={Sinha, Vikas Kumar and Patro, Kiran Kumar and P{\\l}awiak, Pawe{\\l} and Prakash, Allam Jaya}

title={Capsule Network for the Identification of Individuals Using Quantized ECG Signal Images},

author={Prakash, Allam Jaya}

title={Automated detection of myocardial infarction and heart conduction disorders based on feature selection and a deep learning model},

author={Hammad, Mohamed and Chelloug, Samia Allaoua and Alkanhel, Reem and Prakash, Allam Jaya and Muthanna, Ammar and Elgendy, Ibrahim A and P{\\l}awiak, Pawe{\\l}}

title={A hybrid approach of a deep learning technique for real-time ECG beat detection},

author={Kiran Kumar Patro, Allam Jaya Prakash, Saunak Samantray, Joanna P{\\l}awiak, Ryszard Tadeusiewicz, Pawe{\\l} P{\\l}awiak}

title={BAED: A secured biometric authentication system using ECG signal based on deep learning techniques},

author={Prakash, Allam Jaya and Patro, Kiran Kumar and Hammad, Mohamed and Tadeusiewicz, Ryszard and P{\\l}awiak, Pawe{\\l}}

title={A novel approach for brain tumour detection using deep learning based technique},

author={Pedada, Kameswara Rao and Rao, Bhujanga and Patro, Kiran Kumar and Allam, Jaya Prakash and Jamjoom, Mona M and Samee, Nagwan Abdel}

Author response:

We already refer the following research in our manuscript:

A Hybrid Approach of a Deep Learning Technique for Real–Time ECG Beat Detection.

Customized deep learning algorithm for drowsiness detection using single-channel EEG signal.

A novel approach for brain tumour detection using deep learning based technique.

Reviewer #2: This paper presents early detection of breast cancer by DNA sequences using temporal convolutional network. The basic idea sounds good but needs some recommendations to improve the quality of the manuscript.

Concern #1:

Re-write the abstract by significance of your work, limitations of previous works and how your work overcomes those? (state briefly).

Author response:

The abstract has been rewritten:

Early detection of breast cancer can be achieved through mutation detection in DNA sequences, which can be acquired through patient blood samples. Mutation detection can be performed using alignment and machine learning techniques. However, alignment techniques require reference sequences, and machine learning techniques still cannot predict index mutation and require supporting tools. Therefore, in this research, a Temporal Convolutional Network (TCN) model was proposed to detect the type and index mutation faster and without reference sequences and supporting tools. The architecture of the proposed TCN model is specifically designed for sequential labeling tasks on DNA sequence data. This allows for the detection of the mutation type of each nucleotide in the sequence, and if the nucleotide has a mutation, the index mutation can be obtained. The proposed model also uses 2-mers and 3-mers mapping techniques to improve detection performance. Based on the tests that have been carried out, the proposed TCN model can achieve the highest F1-score of 0.9443 for COSMIC dataset and 0.9629 for RSCM dataset, Additionally, the proposed TCN model can detect index mutation six times faster than BiLSTM model. Furthermore, the proposed model can detect type and index mutations based on the patient's DNA sequence, without the need for reference sequences or other additional tools.

Concern #2:

Novelty of the contributions should be highlighted.

Author response:

We describe the novelty of the proposed model in the last paragraph of Introduction section:

The novelty proposed in this paper includes the detection of type and index mutation in DNA sequence data for breast cancer patients using the TCN model. The data inputted is only in the form of DNA sequences and without using reference sequences or other additional tools. The first dataset used was obtained from the public database, namely the Catalogue of Somatic Mutations in Cancer (COSMIC dataset) (22), which includes a combination of the genes ATM, BARD1, BRCA1, BRCA2, BRIP1, CDH1, CHEK2, EPCAM, MLH1, MRE11, MSH2, MSH6, NBN, PALB2, PMS2, PTEN, RAD50, RAD51C, RAD51D, TP53, and XRCC2. Additionally, this research, also use DNA sequence data from breast cancer patients at Cipto Mangunkusumo National Hospital, Jakarta, Indonesia (RSCM dataset) (40). The data is in the form of sequencing results from patient blood samples in the form of short sequences and have a high degree of heterogeneity and with a larger number of sequences.

The architecture of the proposed TCN model is specifically designed for sequential labeling tasks on DNA sequence data. This allows for the detection of the mutation type of each nucleotide in the sequence, and if the nucleotide has a mutation, the index mutation can be obtained. The proposed model can also detect one or more mutations in one input sequence along with the index where the mutation occurs with a faster detection time. 2-mers and 3-mers mapping techniques are also proposed in this research to improve the performance level in detecting type and index mutation. The proposed TCN model succeeded in achieving an F1-score of 0.9427 for the COSMIC dataset and 0.9629 for the RSCM dataset, and was six times faster than BiLSTM model. Fig. 1 shows the advantages and research gap between the proposed TCN model and the related work.

Concern #3:

Better to include a table and summarize previous works along with limitations in the introduction section.

Author response:

We add Fig 1 to summarize the related works and the highlight of proposed TCN model in Introduction section.

Concern #4:

Refer the following important works on importance of CNN

https://doi.org/10.1016/j.bbe.2022.08.004

https://doi.org/10.3390/ijms24010146

http://dx.doi.org/10.34768/amcs-2022-0033

https://doi.org/10.1016/j.acra.2022.12.038

Author response:

We already refer the following research in our manuscript:

https://doi.org/10.1016/j.acra.2022.12.038

https://doi.org/10.3390/ijms24010146

https://doi.org/10.34768/amcs-2022-0033

Concern #5:

Spell out each acronym the first time used in the body of the paper.

Author response:

We already check the manuscript and make sure to spell out each acronym when the first time used in the manuscript.

Concern #6:

Significance of the figure 2 needs to be included.

Author response:

We change the graphical representation of the mutation rate in Fig 3 (Ex Fig 2) to show its significance.

Concern #7:

What is the purpose of Time distributed layer, needs to be addressed.

Author response:

Time distributed layer was used as an output layer because the proposed model is a sequential labelling model, where the length of the output sequence is equal to the input sequence, and with the time distributed layer it can classify each nucleotide in the DNA sequence into a mutated nucleotide label or normal nucleotide.

Concern #8:

Include specific author contributions at the end of the introduction section.

Author response:

We add author contributions at the end of Introduction section:

Finnaly, in this research, all authors planned and conceptualized the conducted research, U.W. carried out the implementation and experimentation of the proposed model, analysis the results, and wrote the draft paper, T.M. led and supervised the project, while A.P. and A.P. analyzed the results and proofread the manuscript.

Concern #9:

Cite the datasets in the reference section. don’t cite like Dataset21.

Author response:

We revised citing Dataset21 to COSMIC dataset and provided the reference to it when we first mentioned COSMIC dataset.

Concern #10:

Cite all the equations, figures and tables in the manuscript.

Author response:

We check the manuscript and make sure all equations, figures, and tables cited.

Concern #11:

Include software and Hardware requirements in the results section.

Author response:

We add the software and hardware used in the results section:

The TCN models were trained using Adam's optimization algorithm (44) in python on Google Colaboratory Pro+.

Concern #12:

In performance parameters calculation why author not much concentrated on accuracy?

Author response:

We use precision, recall, and F1-score metrics to measure model performance because the number of nucleotides in the normal class is far more than the nucleotides in the SNV, insertion, and deletion mutation classes, so the use of accuracy metrics is not suitable for measuring performance.

Concern #13:

A Comparison Table required to compare recent similar works.

Author response:

We add Fig 1 to highlights the comparison between the related works and proposed TCN model in Introduction section.

Concern #14:

Improve the quality of the all figures.

Author response:

We check and refine the low quality of images in our manuscript.

Concern #15:

Read and recheck whole paper and correct grammatical and over spacing errors.

Overall, the idea was good, but needs improvement in the presentation of the work.

Author response:

We already check the whole paper and correct the grammatical error in our manuscript. We hope our manuscript will meet your standard.

---

## [Decision Letter · Decision Letter 1]

7 May 2023

Temporal Convolutional Network for a Fast DNA Mutation Detection in Breast Cancer Data

PONE-D-22-32883R1

Dear Dr. Mengko,

We’re pleased to inform you that your manuscript has been judged scientifically suitable for publication and will be formally accepted for publication once it meets all outstanding technical requirements.

Kind regards,

Mohamed Hammad, Ph.D.

Academic Editor

PLOS ONE

Additional Editor Comments (optional):

Reviewers' comments:

Reviewer's Responses to Questions

**Comments to the Author**

1. If the authors have adequately addressed your comments raised in a previous round of review and you feel that this manuscript is now acceptable for publication, you may indicate that here to bypass the “Comments to the Author” section, enter your conflict of interest statement in the “Confidential to Editor” section, and submit your "Accept" recommendation.

Reviewer #2: All comments have been addressed

2. Is the manuscript technically sound, and do the data support the conclusions?

Reviewer #2: Yes

3. Has the statistical analysis been performed appropriately and rigorously? 

Reviewer #2: N/A

4. Have the authors made all data underlying the findings in their manuscript fully available?

Reviewer #2: Yes

5. Is the manuscript presented in an intelligible fashion and written in standard English?

Reviewer #2: Yes

6. Review Comments to the Author

Reviewer #2: The authors addressed all my comments in the previous round. Check all the figures and Tables citation to the main text. Also proofread with English Native speaker.

7. PLOS authors have the option to publish the peer review history of their article (what does this mean?). If published, this will include your full peer review and any attached files.

Reviewer #2: **Yes: **Kiran Kumar Patro

---

## [Editor Report · Acceptance letter]

17 May 2023

PONE-D-22-32883R1 

*Temporal Convolutional Network for a Fast DNA Mutation Detection in Breast Cancer Data*

Dear Dr. Mengko:

I'm pleased to inform you that your manuscript has been deemed suitable for publication in PLOS ONE. Congratulations! Your manuscript is now with our production department. 

Kind regards, 

on behalf of

Dr. Mohamed Hammad 

Academic Editor

PLOS ONE